# Aldehyde Dehydrogenase 2 (ALDH2) Deficiency, Obesity, and Atrial Fibrillation Susceptibility: Unraveling the Connection

**DOI:** 10.3390/ijms25042186

**Published:** 2024-02-11

**Authors:** Lung-An Hsu, Yung-Hsin Yeh, Chi-Jen Chang, Wei-Jan Chen, Hsin-Yi Tsai, Gwo-Jyh Chang

**Affiliations:** 1Cardiovascular Division, Chang Gung Memorial Hospital, Chang Gung University College of Medicine, Tao-Yuan 33305, Taiwan; yys0tw@yahoo.ca (Y.-H.Y.); chijenformosa@gmail.com (C.-J.C.); wjchen@adm.cgmh.org.tw (W.-J.C.); sunny66house@yahoo.com.tw (H.-Y.T.); 2Graduate Institute of Clinical Medical Sciences, Chang Gung University, Tao-Yuan 33305, Taiwan; gjchang@mail.cgu.edu.tw

**Keywords:** atrial fibrillation, aldehyde dehydrogenase 2, polymorphism, obesity

## Abstract

Atrial fibrillation (AF), characterized by structural remodeling involving atrial myocardial degradation and fibrosis, is linked with obesity and transforming growth factor beta 1 (TGF-β1). Aldehyde dehydrogenase 2 (ALDH2) deficiency, highly prevalent in East Asian people, is paradoxically associated with a lower AF risk. This study investigated the impact of ALDH2 deficiency on diet-induced obesity and AF vulnerability in mice, exploring potential compensatory upregulation of nuclear factor erythroid 2-related factor 2 (Nrf2) and heme-oxygenase 1 (HO-1). Wild-type (WT) and ALDH2*2 knock-in (KI) mice were administered a high-fat diet (HFD) for 16 weeks. Despite heightened levels of reactive oxygen species (ROS) post HFD, the ALDH2*2 KI mice did not exhibit a greater propensity for AF compared to the WT controls. The ALDH2*2 KI mice showed equivalent myofibril degradation in cardiomyocytes compared to WT after chronic HFD consumption, indicating suppressed ALDH2 production in the WT mice. Atrial fibrosis did not proportionally increase with TGF-β1 expression in ALDH2*2 KI mice, suggesting compensatory upregulation of the Nrf2 and HO-1 pathway, attenuating fibrosis. In summary, ALDH2 deficiency did not heighten AF susceptibility in obesity, highlighting Nrf2/HO-1 pathway activation as an adaptive mechanism. Despite limitations, these findings reveal a complex molecular interplay, providing insights into the paradoxical AF–ALDH2 relationship in the setting of obesity.

## 1. Introduction

Atrial fibrillation (AF) is a prevalent cardiac arrhythmia associated with substantial public health implications [1]. AF is characterized by atrial structural and electrical re-modeling, encompassing interstitial fibrosis, myocyte alterations, and shortened atrial refractoriness [2]. On the other hand, obesity has emerged as a global pandemic, independently linked to an increased risk of AF [3]. Weight reduction has been shown to mitigate AF burden [4,5], yet the precise molecular mechanisms underpinning obesity-related AF remain elusive. A growing body of research indicates that obesity contributes to atrial structural remodeling. In animal studies, high-calorie diets induce left atrial enlargement, fibrosis, lipid infiltration, and electrophysiological changes, collectively promoting AF susceptibility [6,7]. Notably, inflammatory processes and fibrosis are implicated as substrates for AF in diet-induced obese animals [8,9,10]. Recent findings have highlighted the importance of specific molecular pathways, notably transforming growth factor beta 1 (TGF-β1), which predominantly enhances the production of type I collagen, in the complex landscape of atrial structural remodeling associated with obesity [11]. Obesity also affects atrial electrical remodeling. Altered ion channel expression and action potential duration have been observed in obese animal models with AF susceptibility [12,13,14]. Oxidative stress and reactive oxygen species (ROS) generation, associated with obesity, may contribute to AF pathogenesis [15]. The intricate interplay between structural and electrical remodeling in obesity-mediated AF remains an active area of investigation. 

Aldehyde dehydrogenase 2 (ALDH2), an enzyme involved in detoxifying aldehydes, is a potential player in obesity-related AF pathogenesis. ALDH2 deficiency, prevalent in East Asian people, has been linked to various cardiovascular conditions, including alcohol sensitivity, myocardial infarction, and ischemic stroke [16,17,18,19]; intriguingly, it seems to confer a paradoxical protection against AF [20,21]. Our recent research indicates that ALDH2 may also play a role in alcohol-related AF by mitigating oxidative stress and aldehyde accumulation, thereby ameliorating myofibril degradation and collagen deposition in the atria [22]. Despite these insights, the intricate relationship between ALDH2 deficiency, obesity, and AF susceptibility remains elusive. This study seeks to investigate the impact of ALDH2 deficiency on diet-induced obesity and AF vulnerability. Additionally, we aim to explore potential compensatory regulatory pathways associated with ALDH2 deficiency, particularly focusing on the upregulation of nuclear factor erythroid 2-related factor 2 (Nrf2) and heme-oxygenase 1 (HO-1). By shedding light on these mechanisms, our research aims to unravel the involvement of ALDH2 in the complex interplay between obesity, oxidative stress, and AF, providing valuable insights into potential therapeutic targets.

## 2. Results

### 2.1. High-Fat Diet Inducing Obesity

The ALDH2*2 knock-in (KI) mutation, created using CRISPR/Cas9 in mouse embryonic stem cells, introduced a G-to-A substitution in exon 12 of the aldh2 locus, mirroring the human ALDH2 (E487K) mutation. This led to reduced ALDH2 enzymatic activity in the liver and decreased protein expression in the heart (Figure 1A). To assess the chronic impact of obesity on AF, wild-type (WT) and homozygous ALDH2*2 KI mice were subjected to either a high-fat diet (HFD: 60% fat, 20% protein, 20% carbohydrates) or a normal diet (ND: 10% fat, 20% protein, 70% carbohydrates) from 8 to 24 weeks of age. Body weights were monitored weekly (Figure 2). After 16 weeks of HFD consumption, both the WT and ALDH2*2 KI mice showed significant weight increases compared to the ND-fed mice. Although the homozygous ALDH2*2 KI mice on the HFD displayed slightly lower mean body weights than their WT counterparts, these differences were not statistically significant. 

### 2.2. 4-HNE and ALDH2 Production in Wild-Type and ALDH2*2 KI Mice with High-Fat Diets

In a previous investigation [22], both heterozygous and homozygous ALDH2*2 KI mice exhibited comparable 4-hydroxy-trans-2-nonenal (4-HNE) production in cardiomyocytes compared to WT mice under baseline conditions. In the current 16-week HFD study, both WT and ALDH2*2 KI mice demonstrated a significant increase in 4-HNE production within cardiomyocytes compared to mice on a normal diet (Figure 1C). Notably, the ALDH2*2 KI mice exhibited 4-HNE production similar to their WT counterparts after prolonged exposure to the HFD (Figure 1C). This phenomenon could be attributed to the marked decrease in ALDH2 production in the hearts of the WT mice following chronic HFD consumption (Figure 1B). Conversely, ALDH2 production in the hearts of the homozygous ALDH2*2 KI mice remained notably low despite prolonged exposure to the HFD. These observations suggest a potential compensatory mechanism in ALDH2*2 KI mice, maintaining 4-HNE production levels similar to WT mice, despite reduced ALDH2 enzyme levels. This intricate interplay between ALDH2 and 4-HNE highlights the impact of genetic mutations on oxidative stress responses in the context of diet-induced obesity. 

### 2.3. AF Inducibility in ALDH2*2 KI Mice with Chronic Diet-Induced Obesity

To assess the susceptibility of WT and homozygous ALDH2*2 KI mice to AF, transesophageal burst pacing was employed following chronic HFD exposure. In line with our previous work, prior to HFD exposure, AF inducibility levels in the ALDH2*2 KI mice mirrored those in their WT counterparts. As depicted in Figure 3, after extended HFD consumption, both groups exhibited significantly increased AF inducibility compared to their same-genotype mice on an ND. Intriguingly, the ALDH2*2 KI mice demonstrated a lower AF inducibility response compared to their WT counterparts following chronic HFD treatment. No significant difference in AF inducibility was observed between the ALDH2*2 KI mice after their prolonged HFD and the WT mice on an ND. Our results suggest that significant differences in AF inducibility are attributed more to the HFD than to the ALDH2*2 genotype. The graphical representation in Figure 3 illustrates the distinctive impact of a chronic HFD on AF susceptibility in ALDH2*2 KI mice compared to WT controls. However, the statistical analysis indicated that the interaction effect between diet-induced obesity and genotype on AF inducibility did not reach significance, underscoring the nuanced relationship between the ALDH2 genotype and chronic HFD exposure in modulating AF vulnerability.

### 2.4. Increased Fat Accumulation and Oxidative Stress in ALDH2*2 KI Mice with Chronic HFD Consumption

To scrutinize the influence of ALDH2*2 on obesity-related oxidative stress, we examined the interplay between ALDH2*2, fat deposition, and oxidative stress within mouse atria subjected to chronic HFD consumption. Notably, our investigation unveiled a significant increase in left atrial fat accumulation for both WT and ALDH2*2 KI mice compared to their respective counterparts on an ND, evident in intense Lipi-Deep red staining (Figure 4A). Crucially, this increase was notably more pronounced in the ALDH2*2 KI mice following prolonged HFD exposure. Moreover, oxidative stress, indicated by heightened ROS generation (Figure 4B), displayed a similar pattern. Both the WT and ALDH2*2 KI mice exposed to the chronic HFD exhibited increased ROS levels in their atria compared to their ND-fed counterparts. Emphasizing this, the rise in ROS levels was more substantial in ALDH2*2 KI mice, accentuating their heightened susceptibility to oxidative stress under diet-induced obesity conditions. This observation underscores increased atrial fat deposition and ROS levels in both WT and ALDH2*2 KI mice following HFD consumption, with the ALDH2*2 KI group showing a more pronounced response compared to their WT counterparts.

### 2.5. Atrial Fibrosis and Structure Remodeling in ALDH2*2 KI Mice with Chronic HFD Consumption

Atrial fibrosis, a key player in AF pathogenesis, is associated with acetaldehyde-induced collagen upregulation [22,23]. In our investigation on mice subjected to a chronic HFD, we explored ALDH2’s potential protective role against collagen expression in the atria. The results revealed a significant upregulation of TGF-β1 in the atria of both ALDH2*2 KI mice and WT controls after chronic HFD consumption, compared to their respective ND-fed counterparts. Notably, the rise in TGF-β1 expression was more pronounced in the ALDH2*2 KI mice than in the WT controls post chronic HFD exposure (Figure 4C). Intriguingly, a contrary trend was observed in atrial fibrosis, with heightened collagen I generation being less pronounced in the atria of ALDH2*2 KI mice compared to WT controls following chronic HFD consumption (Figure 4D). Specifically, atrial fibrosis was more evident in the atria of the WT controls compared to the ALDH2*2 KI mice after prolonged HFD consumption. Similarly, myofibril degradation (indicated by decreased myosin heavy chain (MHC) expression) was more severe in the atria of both the ALDH2*2 KI mice and WT controls after chronic HFD consumption, compared to their respective ND-fed counterparts, while the ALDH2*2 KI mice exhibited myofibril degradation similar to their WT counterparts after prolonged exposure to an HFD (Figure 4E). These findings underscore a complex interplay involving ALDH2, TGF-β1, and collagen I expressions, suggesting a potential modulatory role of ALDH2 in mitigating atrial fibrosis and structure remodeling induced by chronic HFD exposure.

### 2.6. Role of HO-1 in the Differential Effect of Chronic HFD Consumption on ALDH2*2 KI Mice

To unravel the nuanced effects of chronic HFD consumption on AF susceptibility in ALDH2*2 KI mice compared to WT controls, we delved into the potential role of HO-1. Our investigation revealed that, even before HFD consumption, the atria of the ALDH2*2 KI mice exhibited a trend toward a greater expression of HO-1 compared to the WT controls, albeit non-significant (Figure 5A). Following HFD treatment, Nrf2 activation was observed, and this activation was more pronounced and significant in the ALDH2*2 KI mice compared to the WT controls (Figure 5A,B). Acetaldehyde oxidase-derived ROS in the atria of ALDH2*2 KI mice might activate Nrf2, promoting its binding to the antioxidant response element (ARE) region in the HO-1 promoter, ultimately inducing significant HO-1 expression (Figure 5A,B). As demonstrated in our previous study [24], this induction exhibits a protective effect against AF-related remodeling. These findings illuminate an adaptive mechanism that shields cells from escalating oxidative stress, potentially clarifying the differential impact of chronic HFD consumption on AF susceptibility in ALDH2*2 KI mice compared to WT controls. This study sheds light on the intricate interplay between ALDH2, Nrf2, HO-1, and ROS in the context of diet-induced AF vulnerability.

## 3. Discussion

Our investigation reaffirms previous findings indicating that HFD-induced obesity heightens susceptibility to AF in WT mice [7,8]. Strikingly, ALDH2*2 KI mice, despite sharing the obesogenic environment, did not exhibit a greater propensity for AF com-pared to WT controls following chronic HFD treatment. This unexpected resilience in ALDH2*2 KI mice is particularly intriguing, considering the higher ROS levels and lipid accumulation observed in their atria after chronic HFD exposure compared to WT mice. Intriguingly, while the ALDH2*2 KI mice displayed heightened ROS, they produced equivalent levels of 4-HNE and myofibril degradation in their cardiomyocytes compared to their WT counterparts after chronic HFD consumption, indicating suppressed ALDH2 production in WT mice. Moreover, the upregulation of TGF-β1 and collagen I expression in the ALDH2*2 KI mice did not mirror the expected proportional increase seen in a chronic alcohol consumption model [22]. This incongruity prompts an exploration into potential mitigating factors, with a focus on the role of HO-1 as a mediator in this context.

### 3.1. Role of ALDH2 in Obesity-Induced Cardiac Dysfunction

Obesity, induced by an HFD, is intricately linked with heightened ROS generation and subsequent oxidative stress in the body [15]. This oxidative environment has been increasingly associated with the pathogenesis of AF [25]. Moreover, defects in the antioxidant system may contribute to atrial remodeling, with cytotoxic and reactive aldehydes, such as malondialdehyde (MDA) and 4-HNE, emerging as key players in impairing cardiac functions [26,27]. These aldehydes, resulting from lipid peroxidation, form adducts with lipids, proteins, and DNA, leading to their inactivation [28]. Studies, such as Mali et al.’s investigation in mice with metabolic syndrome induced by an HFD, highlight the role of decreased myocardial ALDH2 activity due to 4-HNE adduct formation in contributing to cardiac hypertrophy [29]. In a different approach, Li et al. demonstrated that overexpression of ALDH2 effectively attenuated myocardial remodeling and contractile defects induced by an HFD through the regulation of JNK/AP-1 and IRS-1/Akt signaling pathways [30]. The protective effect of ALDH2 against cardiac remodeling in HFD-induced obesity was further affirmed by Wang et al., who utilized an ALDH2 transgenic mice model [31]. Administration of Alda-1/chaetocin, aimed at enhancing ALDH2 activity, exhibited therapeutic potential by mitigating the impact of palmitic acid on autophagy and contractile function, suggesting a broader role for ALDH2 in obesity-related cardiomyopathy [31]. Our study aligns with these findings, revealing that diet-induced obesity leads to suppressed ALDH2 production, resulting in increased 4-HNE-related oxidative stress and heightened vulnerability to AF. In obesity, characterized by elevated oxidative stress and ROS accumulation, the demand on ALDH2 for the detoxification of acetaldehyde, a toxic byproduct of metabolism, may increase. This heightened demand on ALDH2 may potentially lead to its deficiency, impacting its protective role against oxidative stress. Several factors, including altered gene expression, insulin resistance, fatty liver disease, disrupted signaling pathways, and nutrient imbalances, may contribute to the nuanced relationship between obesity-related changes in metabolism, inflammatory processes, and the expression and activity of ALDH2. Further studies are warranted to unravel the intricate molecular mechanisms underlying the crosstalk between obesity, ALDH2, and cardiac health, shedding light on potential therapeutic targets for obesity-related cardiac complications.

### 3.2. Impact of ALDH2*2 on Body Weight and Metabolic Parameters in Obesity

Our study revealed intriguing observations regarding the body weight of homozygous ALDH2*2 KI mice subjected to an HFD. Notably, despite a visual trend towards lower body weights compared to their WT counterparts on the same diet, these differences did not achieve statistical significance. This contrasts with a recent report indicating that ALDH2*2 homozygous KI male mice are predisposed to diet-induced obesity, presenting with glucose intolerance, insulin resistance, and fatty liver when exposed to a high-fat high-sucrose diet [32]. The underlying mechanisms suggested in this study involve reduced fatty acid oxidation rates and mitochondrial electron transport activity due to increased 4-HNE-adducted proteins in the brown adipose tissue of ALDH2*2 KI mice, leading to decreased thermogenesis and energy expenditure [32]. Interestingly, our findings appear more aligned with human genetic studies than with the reported outcomes in ALDH2*2 KI mice. Human genome-wide association studies (GWASs) and case–control investigations have highlighted that the ALDH2*1 WT allele, rather than the ALDH2*2 allele, is associated with a higher predisposition to metabolic syndrome, hypertension, diabetes, and obesity, particularly in males or individuals with alcohol consumption habits [33,34]. These parallels between our study and human genetic associations underscore the potential translational relevance of our findings. However, it is crucial to acknowledge the limitations of our study, particularly the absence of examinations related to sugar metabolism, insulin resistance, lipid profiles, and blood pressure. Further comprehensive investigations are warranted to unravel the intricate interplay between ALDH2*2, diet-induced obesity, and the broader spectrum of metabolic parameters. Such endeavors will enhance our understanding of the clinical implications of ALDH2*2 in obesity-related metabolic disturbances, paving the way for more-targeted therapeutic strategies.

### 3.3. Expanding beyond Alcohol Metabolism: ALDH2*2 in Obesity-Related Cardiovascular Complications

Traditionally, the research focus on ALDH2*2 has centered on its implications in alcohol metabolism, linking it to increased risks of alcohol-related cancers, cardiovascular diseases, and alcohol use disorders. This heightened risk is attributed to its role in diminishing acetaldehyde metabolism efficiency. However, our understanding of the genetic variant has evolved, transcending its conventional role. GWASs and phenome-wide association studies have underscored its intriguing potential to decouple obesity from associated health complications [35,36]. In a study by Hu et al., the impact of ALDH2*2 on the osteogenic and adipogenic differentiation of 3T3-L1 preadipocytes was investigated [36]. Notably, ALDH2-WT cells exhibited significantly higher collagen type I mRNA expression and more mineralized nodules than control cells or those expressing ALDH2*2. Aligning with these findings, our study observed less-pronounced atrial fibrosis, reflected by collagen production in the atria of ALDH2*2 KI mice compared to wild-type controls following chronic HFD consumption. While acetaldehyde has been recognized for its fibrogenic role in upregulating the transcription of collagen I directly and indirectly inducing the synthesis of TGF-β1 in alcohol-induced hepatic fibrosis [23], our previous study demonstrated that ALDH2*2 KI mice subjected to chronic alcohol intoxication exhibited a higher degree of 4-HNE accumulation, increased TGF-β1 expression, and enhanced collagen deposition in their atria compared to wild-type mice [22]. Importantly, these effects were mitigated by the ALDH2-selective activator, Alda-1 [22]. Contrary to the pronounced increase in ROS and TGF-β1 in the atria of ALDH2*2 KI mice following chronic HFD consumption, our study revealed that compensatory upregulation of the Nrf2 and HO-1 pathway might attenuate atrial fibrosis in these mice. In the context of ALDH2 deficiency, ROS triggers the inactivation of the E3 ubiquitin ligase within the Keap1 complex, preventing Nrf2 ubiquitination and inhibiting its degradation by the proteasome, ultimately resulting in heightened Nrf2 protein expression [37]. This identified signaling pathway likely serves as the basis for the compensatory upregulation of the Nrf2 and HO-1 pathway observed in our study, offering insight into the intricate regulatory dynamics in response to ALDH2 deficiency and chronic HFD consumption. This observation aligns with the lower degree of AF inducibility response demonstrated by ALDH2*2 KI mice compared to their wild-type counterparts following chronic HFD treatment. Human studies have further hinted at a potential beneficial effect of ALDH2*2 on AF [20,21]. There might be additional compensatory regulatory pathways associated with ALDH2 deficiency to counteract susceptibility to obesity-related AF. Recent research has implicated the short-chain fatty acid propionate in activating free fatty acid receptor-3, a process frequently observed in obesity, leading to inflammation and fibrosis in cardiac cells. Notably, the regulator of G-protein signaling, (RGS)-4, acts as a ‘molecular brake’, counteracting these effects and modulating parasympathetic signaling. It regulates heart rate and suppresses arrhythmogenic calcium signaling [38,39]. This underscores a broader cardioprotective potential against AF pathogenesis, indicating a promising avenue for further research into its role in AF development within the context of obesity and ALDH2 deficiency. Our findings unravel a complex interplay between the ALDH2*2 genetic variant, obesity, and associated complications such as AF. This expands the narrative surrounding ALDH2*2, emphasizing its role not only in alcohol-related health risks but also in the broader context of cardiovascular remodeling influenced by obesity.

### 3.4. Limitations

Our study has limitations, including not examining insulin resistance, lipid profiles, and blood pressure in our mouse model. Although our specific pacing protocol demonstrated reproducibility in assessing AF inducibility, alternative pacing protocols may yield different results. Previous GWASs have suggested that ALDH2*2’s association with obesity and cardiovascular risk factors may be influenced by alcohol consumption habits [33,34]. Future investigations combining an HFD and chronic alcohol consumption in our mouse model, along with a broader spectrum of pacing approaches, could enhance understanding of AF susceptibility in obesity with ALDH2 deficiency. Additionally, we did not elucidate the detailed signaling mechanisms explaining the reduced AF inducibility in ALDH2*2 mice with obesity. Further studies, incorporating multi-omics and gut microbiota analyses, are needed to untangle ALDH2*2′s decoupling of excess AF inducibility from obesity. Finally, our study did not explore the potential role of electrical remodeling in AF susceptibility within the context of ALDH2 deficiency and diet-induced obesity. Addressing these limitations in future research will contribute to a more comprehensive understanding of the intricate relationship between ALDH2 deficiency, obesity, and AF susceptibility. 

## 4. Materials and Methods

### 4.1. Ethics Statement

All animal experimental procedures were approved by the Institutional Animal Care and Use Committee of Chang Gung University (Taoyuan, Taiwan; IACUC No. CGU110-006, 12 May 2021), and the experiments were conducted following the relevant guidelines.

### 4.2. Generation of ALDH2*2 KI Mice Using CRISPR/CAS9 to Mimic Humans with ALDH2*2

ALDH2 is highly conserved in mice and humans. To replicate the ALDH2*2 variant found in humans, we employed a CRISPR/Cas9 system to generate an ALDH2*2 KI mouse model on a C57BL/6J background. This model involved introducing a single nucleotide substitution (G to A) within exon 12 of the aldh2 genomic fragment, precisely mimicking the position of the human E487K mutation. This inactivating point mutation was designed to simulate the genetic alteration associated with ALDH2*2, allowing for a detailed investigation of its effects in a murine context. The methodology for creating this model was consistent with established procedures, as outlined in previous research [22]. This approach ensured the accurate representation of the ALDH2*2 mutation in mice, providing a valuable tool for studying the impact of this genetic variant on various physiological processes.

### 4.3. Mice Maintenance and Diet-Induced Obesity Model

ALDH2*2 KI mice were bred and maintained in a barrier facility under pathogen-free conditions at Chang Gung University. Wild-type littermates of the ALDH2*2 KI mice served as controls for all analyses. The mice were housed on a 12 h light/12 h dark cycle and provided with food and water ad libitum. The mice were older than 8 weeks at the beginning of the experiments. A chronic diet-induced obesity model was established by administering a high-fat diet (HFD; containing 60 kcal.% fat, 20 kcal.% protein, and 20 kcal.% carbohydrates) compared to a normal diet (ND; containing 10 kcal.% fat, 20 kcal.% protein, and 70 kcal.% carbohydrates) at the age of 8 weeks to 16 weeks. Body weights were recorded weekly.

### 4.4. Programmed Electrical Stimulation to Evaluate AF Vulnerability

Transesophageal stimulation, conducted in accordance with established procedures [40] and featuring adjustments from decremental atrial pacing to burst atrial pacing at a fixed cycle length [24], was conducted to assess AF vulnerability. The mice were anesthetized with Zoletil (50mg/kg) and Xylazine (10mg/kg) intraperitoneally. Utilizing a 4F electrode catheter (ST. JUDE MEDICAL, St. Paul, MN, USA) inserted into the esophagus, connected to an isolated stimulator (SI-200, iWork Systems Inc., WA, USA), atrial pacing was performed. IX-TA-220 and LabScribe software v3 (iWork Systems Inc., WA, USA) managed the pacing programs and EKG recordings. Pacing parameters included an amplitude of 1.5x diastolic capture threshold and a duration of 2 ms. A pre-test burst ensured atrial stimulation capture, followed by 10 repeats of pacing bursts at a cycle frequency of 30 Hz for 3 s. AF, defined as a period of rapid irregular atrial rhythm lasting over 3 s, was assessed. AF inducibility was expressed as the ratio of pacing-triggered AF episodes to 10 pacing bursts in each mouse.

### 4.5. Western Blot Analysis to Study ALDH2-Related Oxidative Stress and Atrial Remodeling

Equal amounts of proteins were extracted from tissue and subjected to sodium dodecyl sulfate-polyacrylamide gel electrophoresis. After transferring them to nitrocellulose membranes (Perkin Elmer, Waltham, MA, USA), the proteins were incubated with primary antibodies against ALDH2, 4-HNE, HO-1 (Abcam, Cambridge, UK), Nrf2, (ABclonal, MA, USA), tubulin and GAPDH (Santa Cruz, TX, USA). Signals were detected with electrochemiluminescence (Santa Cruz, TX, USA) and quantified by densitometry. Data in the linear immunoreactive range were normalized to GAPDH or α-tubulin as a loading control.

### 4.6. Histology and Immunohistochemistry Analyses to Examine Atrial ALDH2-Related Oxidative Stress and Atrial Remodeling

Mouse atrial tissues, embedded in O.C.T compound (Sakura Finetek, St. Torrance, CA, USA), were sectioned into five-micrometer cross-sections. Lipid accumulation in the left atrium myocardium was examined using Lipi-Deep red staining (LD-04, Dojindo Laboratories, Kumamoto, Japan). Confocal microscopy, employing primary antibodies against MHC, TGF-β1, and collagen I (Abcam, Cambridge, UK, Delaware Avenue, CA, and Santa Cruz, TX, USA, respectively), followed by FITC or Cy3-conjugated secondary antibodies (Abcam), facilitated immunohistochemical and cytochemical analyses. Nuclei were visualized through DAPI staining. Protein expression levels were quantified by calculating the protein-occupied area in the tissue divided by the nuclear area. ROS levels in the atria were assessed using the fluorescent dye dihydroethidium. Tissue samples were pre-incubated with 10 μmol/L dihydroethidium for 30 min at room temperature, and ROS-mediated fluorescence was observed under a confocal microscope (Leica TCS SP8, Wetzlar, Germany). Excitation at 518 nm using an argon laser and emission recording (>600 nm) allowed the acquisition of two-dimensional images (512 × 512 pixels).

### 4.7. Real-Time Quantitative Reverse Transcription–PCR (RT-PCR)

Total cellular RNA was extracted from tissues using the TRIzol reagent (Life Technologies, Rockville, MD, USA) and real-time quantitative RT-PCR was performed as described previously [41]. GAPDH mRNA was used as an internal control. The primers were listed as follows: mNrf2: forward: 5′-CTGAACTCCTGGACGGGACTA-3′ and reverse: 5′- CGGTGGGTCTCCGTAAATGG-3′; mHO-1: forward: 5′-CACTCTGGAGATGACACCTGAG-3′ and reverse: 5′-GTGTTCCTCTGTCAGCATCACC-3′; mGAPDH: forward: 5′-CGACTTCAACAGCAACTCCCACTCTTCC-3′ and reverse: 5′-TGGGTGGTCCAGGGTTTCTTACTCCTT-3′. Relative expressions of Nrf2 and HO-1 were calculated using the 2^−ΔΔct^ method via the SYBR green detection mechanism.

### 4.8. Statistical Analysis 

Continuous variables, expressed as their mean ± SD, were assessed for normal distribution using the Kolmogorov–Smirnov test. An independent Student’s *t*-test and one-way ANOVA with post hoc Tukey’s test were applied for the two groups and multiple comparisons, respectively. Interactions between diet and genotype in relation to AF inducibility were tested with a two-way ANOVA. A *p* value of <0.05, using the two-way test, was considered statistically significant. SPSS software, version 20.0 (SPSS Inc., Chicago, IL, USA), performed all statistical analyses.

## 5. Conclusions

Our study on ALDH2 deficiency and pacing-induced AF in a murine model treated with a chronic HFD revealed unexpected insights. Contrary to expectations, ALDH2 deficiency did not significantly heighten AF susceptibility in obesity; instead, the activation of the Nrf2/HO-1 pathway suggests an adaptive mechanism. Additionally, our results indicate that significant differences in AF inducibility were primarily attributed to the HFD rather than the ALDH2*2 genotype. These unexpected findings emphasize the need for further investigation into the nuanced relationships between ALDH2 deficiency, obesity, and AF susceptibility. Such insights may guide targeted treatments, especially in populations with a high prevalence of the ALDH2*2 allele.

## Figures and Tables

**Figure 1 ijms-25-02186-f001:**
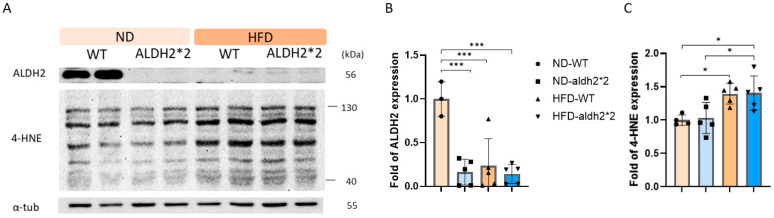
(**A**) Representative Western blot and quantification relative to α-tubulin of (**B**) aldehyde dehydrogenase 2 (ALDH2) and (**C**) 4-hydroxy-trans-2-nonenal (4-HNE) production in the heart of wild-type (WT) and homozygous ALDH2*2 KI mice treated with either a normal diet (ND) or high-fat-diet (HFD) for 16 weeks. * *p* < 0.05, *** *p* < 0.001.

**Figure 2 ijms-25-02186-f002:**
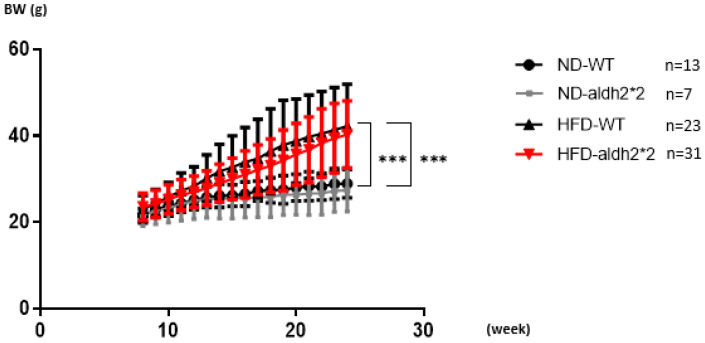
Body weight (BW) change of wild-type (WT) and homozygous ALDH2*2 KI mice treated with either a normal diet (ND) or high-fat-diet (HFD) for 16 weeks. *** *p* < 0.0001: HFD-WT vs. ND-WT and HFD-ALDH2*2 KI vs. ND-ALDH2*2; no significant difference: HFD-WT vs. HFD-ALDH2*2.

**Figure 3 ijms-25-02186-f003:**
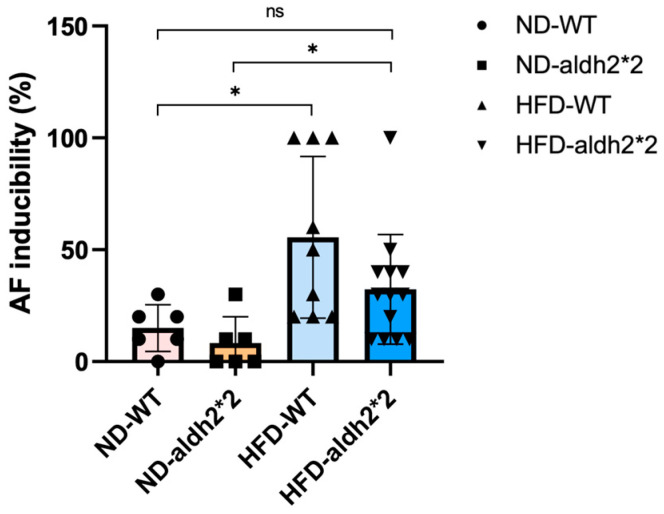
Impact of transesophageal burst pacing on atrial fibrillation (AF) occurrence in wild-type (WT) and ALDH2*2 KI mice consuming either a normal diet (ND) or chronic high-fat-diet (HFD) for 16 weeks. *: *p* < 0.05; ns: non-significant (*p* > 0.05).

**Figure 4 ijms-25-02186-f004:**
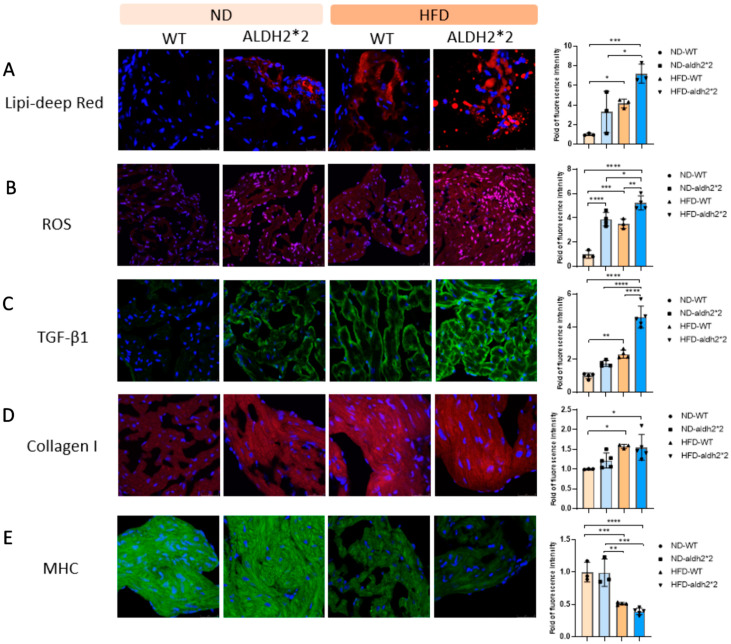
Representative confocal images: (**A**) fat deposition (Lipi-Deep red), (**B**) reactive oxygen species (ROS), (**C**) transforming growth factor beta 1 (TGF-β1), (**D**) collagen I, and (**E**) myosin heavy chain (MHC) production in the atria of wild-type (WT) and homozygous ALDH2*2 KI mice treated with either a normal diet (ND) or a high-fat diet (HFD) for 16 weeks. Quantification of relative fluorescence density is shown on the right. A minimum of 10 random fields were selected for scanning and averaging; data are presented as the mean ± SD. * *p* < 0.05, ** *p* < 0.01, *** *p* < 0.001, **** *p* < 0.0001.

**Figure 5 ijms-25-02186-f005:**
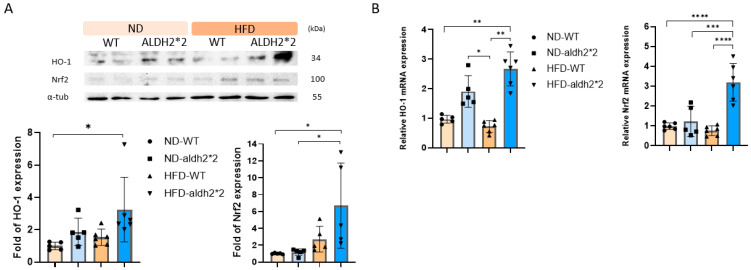
(**A**) Representative Western blot and quantification relative to α-tubulin of heme oxygenase-1 (HO-1) and nuclear factor erythroid 2-related factor 2 (Nrf2) production in the heart of wild-type (WT) and homozygous ALDH2*2 KI mice treated with either a normal diet (ND) or a high-fat-diet (HFD) for 16 weeks. (**B**) Relative fold change of mRNA expression of Nrf2 and HO-1 in the heart of WT and homozygous ALDH2*2 KI mice treated with either an ND or HFD for 16 weeks, as measured with real-time quantitative PCR. * *p* < 0.05, ** *p* < 0.01, *** *p* < 0.001, **** *p* < 0.0001.

## Data Availability

Data are contained within the article.

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
