# Peer review of "Aldehyde Dehydrogenase 2 (ALDH2) Deficiency, Obesity, and Atrial Fibrillation Susceptibility: Unraveling the Connection"

_ijms, 2024, doi:10.3390/ijms25042186_

Round 1

Reviewer 1 Report

Comments and Suggestions for Authors

Lung Han Su et al. evaluated in this study the impact of aldehyde dehydrogenase 2 (ALDH2) deficiency on atrial fibrillation (AF) vulnerability induced by high-fat-diet (HFD)-related obesity with a murine model with ALDH2*2 knock-in (KI) mice generated by CRISPR/CAS9 proving ALDH2 deficiency's role in HFD induced AF vulnerability.

The introduction is comprehensive and the methods are clearly presented. The results are clear and the discussions are well written and support the conclusions.

Author Response

 We greatly appreciate the recommendations of the reviewer.

Reviewer 2 Report

Comments and Suggestions for Authors

Dear Authors,

the risk factors for AF are quite interesting to research as this disease incidence is increasing and likely continues to do. This study by Hsu et al. ”explores the impact of aldehyde dehydrogenase 2 (ALDH2) deficiency on atrial fibrillation (AF) vulnerability induced by high-fat-diet (HFD)-related obesity” in a murine model. (lines 14f).

There are some concerns.

Abstract. Please be sure to explain every abbreviation before 1st usage. The reason for measuring myofibril degeneration, tgf, nrf2 and ho is not clear. The results are not clearly presented in regard to the aim of the study.

Introduction. What about inflammation (line 41) or chad11? you did not (aim to) measure it. instead, this section should be used to explain the interplay of the things you measured (see my comment to the abstract). your study is about aldh, so it probably would be a good idea to go a little deeper into this also.

Results. Fig1 nicely shows, that aldh is not expressed in the aldh2*2 groups. and, however, in the WT-HFD group; can you explain that? In line 65 you mention heart, but it should mean atria, shouldn’t it? Btw which, left or right, or both? Fig2 * usually are in the figure, not the legend. Self-citing is not necessary here (line100). Fig3 shows the data, but I could not find numbers or a p value between to the groups of interest: ND-WT and HFD-ALDH2*2. The data tell, as you describe correctly (lines 20, 103) that AF inducibility is similar in HFD-aldh2*2 to in HFD-WT; and in ND-ALDH2*2 to ND-WT. This suggests a role for HFD, but not for aldh.

Discussion. Thus, you are correct saying that the “investigation reaffirms previous findings indicating that HFD-induced obesity heightens susceptibility to AF” (line 175). Nrf2 is a negative regulator of inflammation, thus not a good measure for inflammation itself.

Methods. Which adjustments (line 300)? please use +-SD instead of sem.

Conclusion. As mentioned above, your data do not support a difference between WT and ALDH2*2. “Tachy” is mentioned 1st time in the conclusion.

Further, the switch between different font styles makes it not easier to read. A scheme would help to better understand, what you aim to investigate.

Comments on the Quality of English Language

Some sentences are really hard to understand. 

Author Response

We greatly appreciate the recommendations of the reviewer.

Abstract. Please be sure to explain every abbreviation before 1st usage. The reason for measuring myofibril degeneration, tgf, nrf2 and ho is not clear. The results are not clearly presented in regard to the aim of the study.

Response: In accordance with the reviewer's feedback, we have meticulously addressed the concerns raised in the Abstract. Specifically, we have ensured that every abbreviation is explained upon its initial usage to enhance clarity for the readers. Additionally, we have revised the Abstract to provide a more explicit rationale for measuring myofibril degradation, TGF-beta 1, Nrf2, and HO-1. The revised content now better elucidates the relevance of these factors to the focus of our study, fostering a clearer understanding of the objectives and outcomes. We appreciate the reviewer's input and believe these modifications contribute to the overall improvement of the Abstract.

Introduction. What about inflammation (line 41) or chad11? you did not (aim to) measure it. instead, this section should be used to explain the interplay of the things you measured (see my comment to the abstract). your study is about aldh, so it probably would be a good idea to go a little deeper into this also.

Response: Acknowledging the reviewer's valuable feedback, we have made significant revisions to the Introduction section. Specifically, we have provided a more comprehensive explanation for measuring myofibril degradation, TGF-beta, Nrf2, and HO-1, emphasizing their interconnectedness with ALDH2—the primary focus of our study. This adjustment aims to offer a clearer context for the chosen parameters and their relevance to the objectives of our research. We appreciate the reviewer's insightful comments, and we believe that these refinements contribute to a more cohesive and informative Introduction section.

Results. Fig1 nicely shows, that aldh is not expressed in the aldh2*2 groups. and, however, in the WT-HFD group; can you explain that? In line 65 you mention heart, but it should mean atria, shouldn’t it? Btw which, left or right, or both? Fig2 * usually are in the figure, not the legend. Self-citing is not necessary here (line100). Fig3 shows the data, but I could not find numbers or a p value between to the groups of interest: ND-WT and HFD-ALDH2*2. The data tell, as you describe correctly (lines 20, 103) that AF inducibility is similar in HFD-aldh2*2 to in HFD-WT; and in ND-ALDH2*2 to ND-WT. This suggests a role for HFD, but not for aldh.

Response: (1) In the WT-HFD group, while the figure may suggest no expression of ALDH2, it is more accurately described as a decrease in expression. The faint band may be obscured during the figure processing, and this has been amended. (2) Regarding the mention of 'heart' in line 65, the protein extraction is indeed from heart tissue, and the confocal study specifically pertains to the left atria. (3) The asterisks in Fig2 have been relocated to the figure, and the unnecessary self-citing in the original line 100 has been removed. (4) In response to your observation regarding the absence of p-values between the ND-WT and HFD-ALDH2*2 groups in Fig 3, we have added “ns, nonsignificant (p-value > 0.05)” to indicate the lack of statistically significant differences between these groups. We appreciate your attention to this crucial aspect of our data interpretation.

Discussion. Thus, you are correct saying that the “investigation reaffirms previous findings indicating that HFD-induced obesity heightens susceptibility to AF” (line 175). Nrf2 is a negative regulator of inflammation, thus not a good measure for inflammation itself.

Response: We appreciate the reviewer's insightful observation. In response, we have clarified the rationale behind measuring the effect of Nrf2 in the revised Introduction. In our study, we sought to explore the potential compensatory regulatory pathway represented by Nrf2/HO-1, given the seemingly paradoxical protection against AF associated with ALDH2 deficiency in human studies. While acknowledging that Nrf2 is a known negative regulator of inflammation, we recognize that it may not directly measure inflammation itself. Our focus on Nrf2 is grounded in its potential role within the intricate network associated with ALDH2 deficiency.

Methods. Which adjustments (line 300)? please use +-SD instead of sem.

Response: We have incorporated details regarding the adjustments made from decremental atrial pacing to burst atrial pacing at a fixed cycle length in the revised Methods section. Additionally, we have modified our data presentation in figures to use ± SD instead of SEM, as suggested by the reviewer.

Conclusion. As mentioned above, your data do not support a difference between WT and ALDH2*2. “Tachy” is mentioned 1st time in the conclusion.

Response: In response to the reviewer's observation, we have revised our conclusion for clarity. Our study investigated the relationship between ALDH2 deficiency and susceptibility to pacing-induced AF in a murine model subjected to chronic high-fat diet, characterized by oxidative stress and atrial structural remodeling. Despite not observing a significant difference between WT and ALDH2*2, our findings still highlight the potential implications of ALDH2 deficiency in the context of chronic HFD-induced AF. The term “Tachy” has been removed to simply the text.

 Further, the switch between different font styles makes it not easier to read. A scheme would help to better understand, what you aim to investigate.

Response: In response to your feedback, we have revised the formatting by removing the italic text to ensure a consistent and more readable presentation. In the revised Introduction, results, and Figures, we have refrained from including a scheme, as we believe they now clearly convey our aims without the need for additional visual aids. Thank you for your suggestion, and we hope these adjustments enhance the overall clarity of the manuscript.

Some sentences are really hard to understand. 

Response: In our revisions, we have made efforts to improve the overall readability and comprehension of the text.

Reviewer 3 Report

Comments and Suggestions for Authors

The paper of Hsu and coll. is very interesting, but some minor points should be clarified:

-       Could the results of atrial fibrillation susceptibility be different in the case of a different pacing protocol?

-       In the first paragraph of discussion the authors state that: “Strikingly, ALDH2*2 KI mice, despite sharing the obesogenic environment, did not exhibit a greater propensity for AF compared to WT controls following chronic HFD treatment.”. But, in the conclusion, they affirm: “Our study establishes a connection between ALDH2 deficiency and heightened susceptibility to tachypacing-induced AF in a murine model subjected to chronic high-fat diet, characterized by oxidative stress and atrial structural remodeling”. The two sentences seem in contrast.

-       In the opinion of the authors, could a rat model, in which high fat diet and excessive alcohol consumption are both tested, give different results?

Author Response

We greatly appreciate the recommendations of the reviewer.

-       Could the results of atrial fibrillation susceptibility be different in the case of a different pacing protocol?

Response: Thank you for highlighting the potential influence of different pacing protocols on atrial fibrillation susceptibility. In the Limitation section of our revised manuscript, we acknowledge the variability associated with various transesophageal atrial pacing approaches. Our selected pacing protocol, detailed in previous works (ref 22 and 24), has consistently demonstrated reproducibility in assessing atrial fibrillation inducibility.

-       In the first paragraph of discussion the authors state that: “Strikingly, ALDH2*2 KI mice, despite sharing the obesogenic environment, did not exhibit a greater propensity for AF compared to WT controls following chronic HFD treatment.”. But, in the conclusion, they affirm: “Our study establishes a connection between ALDH2 deficiency and heightened susceptibility to tachypacing-induced AF in a murine model subjected to chronic high-fat diet, characterized by oxidative stress and atrial structural remodeling”. The two sentences seem in contrast.

Response: In response to the reviewer's observation, we have revised our conclusion for clarity. Our study investigated the relationship between ALDH2 deficiency and susceptibility to pacing-induced AF in a murine model subjected to chronic high-fat diet, characterized by oxidative stress and atrial structural remodeling. Despite not observing a significant difference between WT and ALDH2*2, our findings still highlight the potential implications of ALDH2 deficiency in the context of chronic HFD-induced AF. 

-       In the opinion of the authors, could a rat model, in which high fat diet and excessive alcohol consumption are both tested, give different results?

Response: We appreciate the insightful comment from the reviewer. Indeed, we concur that exploring the combined effects of a high-fat diet and chronic alcohol consumption in a rat model may yield different results. This intriguing avenue presents an interesting prospect for future investigations, and we plan to undertake studies specifically designed to address this aspect, which is shown in the Limitation section of our revised manuscript. 

Reviewer 4 Report

Comments and Suggestions for Authors

1) The data on Nrf2 and HO-1 upregulation is not convincing at all. The authors need to repeat thei western blots more rigorously and/or perform real time PCR to show mRNA upregulation.

2) The authors need to discuss potential signaling mechanisms underlying their findings.

3) In the same vein, one possible mechanism could be via regulation of RGS protein levels, particularly RGS4, which is known to underlie AFib pathogenesis risk (see: PMID: 35628613; PMID: 37724539). The authors could perhaps speculate on this in "Discussion". 

4) The limitations of the study need to be listed in "Discussion".

Comments on the Quality of English Language

Minor editing required.

Author Response

We greatly appreciate the recommendations of the reviewer.

1) The data on Nrf2 and HO-1 upregulation is not convincing at all. The authors need to repeat the western blots more rigorously and/or perform real time PCR to show mRNA upregulation.

Response: In response to the reviewer's suggestion, we have conducted additional experiments to enhance the robustness of our data. Specifically, we have repeated the Nrf2 western blots with increased rigor. Additionally, we have performed real-time PCR analysis to complement our findings, providing further evidence for the upregulation of Nrf2 and HO-1 mRNA expression. The results of these additional analyses strengthen the validity and reliability of our conclusions regarding Nrf2 and HO-1 upregulation.

2) The authors need to discuss potential signaling mechanisms underlying their findings.

Response: In accordance with the reviewer’s suggestion, we have added a potential signaling mechanisms underlying our findings as follows: In the context of ALDH2 deficiency, ROS triggers the inactivation of the E3 ubiquitin ligase within the Keap1 complex, preventing Nrf2 ubiquitination and inhibiting its degradation by the proteasome, ultimately resulting in heightened Nrf2 protein expression [37]. This identified signaling pathway likely serves as the basis for the compensatory upregulation of the Nrf2 and HO-1 pathway observed in our study, offering insight into the intricate regulatory dynamics in response to ALDH2 deficiency and chronic high-fat diet consumption.

3) In the same vein, one possible mechanism could be via regulation of RGS protein levels, particularly RGS4, which is known to underlie AFib pathogenesis risk (see: PMID: 35628613; PMID: 37724539). The authors could perhaps speculate on this in "Discussion". 

Response: In response to the reviewer's valuable suggestion, we have included speculation in the revised Discussion regarding the potential involvement of RGS4 in the mechanism underlying AFib pathogenesis risk. We have cited relevant literature (PMID: 35628613; PMID: 37724539) to support this speculation, emphasizing the need for further research to explore and elucidate the role of RGS proteins, particularly RGS4, in the context of our study. This addition aims to stimulate future investigations and discussions on the intricate molecular mechanisms contributing to AFib susceptibility in the context of ALDH2 deficiency and diet-induced obesity.

4) The limitations of the study need to be listed in "Discussion".

Response: As recommended by the reviewer, we have incorporated a new paragraph to discuss the limitations of our study. In this section, we address key constraints such as [not examining insulin resistance, lipid profiles, and blood pressure; potential influence of alcohol consumption habits on ALDH2*2's association with obesity and cardiovascular risk factors; reliance on a specific pacing protocol with potential differences in alternative pacing approaches; lack of elucidation of detailed signaling mechanisms explaining reduced AF inducibility in ALDH2*2 with obesity; need for future studies incorporating multi-omics and gut microbiota analysis; and absence of exploration of electrical remodeling in AF susceptibility within the context of ALDH2 deficiency and diet-induced obesity], aiming to provide a transparent overview of the challenges and considerations associated with our research.

Reviewer 5 Report

Comments and Suggestions for Authors

I have received for review a case control study entitled “Aldehyde Dehydrogenase 2 (ALDH2) Deficiency, Obesity, and Atrial Fibrillation Susceptibility: Unraveling the Connection” which is being processed for publication in the journal Internatiomal Journal of Molecular Sciences.

The manuscript proposed by the authors is an extremely interesting one which provides new data on the pathophysiology of atrial fibrillation and its links to obesity.

I would like to congratulate the collective of authors for the proposed manuscript. The medical information presented is of a high scientific quality, but I believe that the authors should address the following issues:

·       Lines 21-24 - does the text need to be formatted in italics?

·       Introduction - I suggest presenting more extensively the implications of ALDH2 in cardiovascular pathologies and clarifying the limited data to date in relation to atrial fibrillation.

·       Results - many paragraphs have italic text, is it necessary to format the text in italics? Insert a table with the data obtained to complement the figures.

·       Discussion - same issue regarding text formatting

·       Insert a paragraph on the limitation of the study.

Conclusions: Well structured, in accordance with the results presented.

In conclusion, the proposed manuscript brings to attention an extremely interesting topic, presenting scientific information which offers future research directions. The quality of the manuscript will be improved if the authors take into account the remarks made above.

Author Response

We greatly appreciate the recommendations of the reviewer.

  • Lines 21-24 - does the text need to be formatted in italics?

Response: We have revised the formatting, and the italic text has been removed.

  • Introduction - I suggest presenting more extensively the implications of ALDH2 in cardiovascular pathologies and clarifying the limited data to date in relation to atrial fibrillation.

Response: In accordance with the reviewer’s suggestion, we have revised our introduction by acknowledging the intriguing paradoxical protection against AF conferred by ALDH2 deficiency. This addresses the reviewer's concern about the limited data on this aspect.

  • Results - many paragraphs have italic text, is it necessary to format the text in italics? Insert a table with the data obtained to complement the figures.

Response: We have revised the formatting, and the italic text has been removed. While we appreciate the suggestion to insert a table for every figure, we believe that it might appear redundant given the comprehensive presentation of data in the figures. However, we are open to further discussion and can certainly consider additional tables if deemed necessary for a clearer presentation of the results.

  • Discussion - same issue regarding text formatting

Response: We have revised the formatting, and the italic text has been removed.

  • Insert a paragraph on the limitation of the study.

Response: As recommended by the reviewer, we have incorporated a new paragraph to discuss the limitations of our study. In this section, we address key constraints such as [not examining insulin resistance, lipid profiles, and blood pressure; potential influence of alcohol consumption habits on ALDH2*2's association with obesity and cardiovascular risk factors; reliance on a specific pacing protocol with potential differences in alternative pacing approaches; lack of elucidation of detailed signaling mechanisms explaining reduced AF inducibility in ALDH2*2 with obesity; need for future studies incorporating multi-omics and gut microbiota analysis; and absence of exploration of electrical remodeling in AF susceptibility within the context of ALDH2 deficiency and diet-induced obesity], aiming to provide a transparent overview of the challenges and considerations associated with our research.

Conclusions: Well structured, in accordance with the results presented.

 In conclusion, the proposed manuscript brings to attention an extremely interesting topic, presenting scientific information which offers future research directions. The quality of the manuscript will be improved if the authors take into account the remarks made above.

Response: We greatly appreciate the recommendations of the reviewer.

Round 2

Reviewer 2 Report

Comments and Suggestions for Authors

Dear Authors,

This study investigated the impact of ALDH2 deficiency on diet-induced obesity and AF vulnerability in mice, exploring potential compensatory upregulation of nuclear factor erythroid 2-related factor 2 (Nrf2) and heme-oxygenase 1 (HO-1). (lines 17f)

Figure 2 tells that there is no difference/body weight in aldh2*2 vs wt. hfd leads to differences in body weight. in this figure TT stands for aldh2*2

Figure 3 shows that there is no difference/AF inducibility in aldh2*2 vs wt. again, hfd leads to the differences. here groups are presented in a different order compared to other figures.

Figure 5 shows an increased expression of HO-1 in aldh2*2 vs. wt mRNA and protein, and increased expression of nrf2 in hfd/aldh2*2 (mRNA and protein); however n=3 and SD quite high.

The conclusion is not supported by the presented results. instead, meaninless statements are given.

Although this field of research has interesting content, this study contains methodological and many presentational deficits.

Comments on the Quality of English Language

Scientific English ok. 

Author Response

 We greatly appreciate the recommendations of the reviewer.

Figure 2 tells that there is no difference/body weight in aldh2*2 vs wt. hfd leads to differences in body weight. in this figure TT stands for aldh2*2

Response: We appreciate the reviewer's feedback. To address potential confusion, we have replaced the notation 'TT' with aldh2*2 in the revised Figure 2. Additionally, we have revised the legend to enhance clarity:

Figure 2. Body weight (BW) change in wild-type (WT) and homozygous ALDH2*2 KI mice treated with either a normal diet (ND) or high-fat diet (HFD) for 16 weeks. *** p < 0.0001, HFD-WT vs. ND-WT and HFD-ALDH2*2 KI vs. ND-ALDH2*2; No significant difference HFD-WT vs. HFD-ALDH2*2.

These modifications aim to ensure accurate representation and interpretation of the data. We appreciate the reviewer's valuable input in improving the clarity of our figures.

Figure 3 shows that there is no difference/AF inducibility in aldh2*2 vs wt. again, hfd leads to the differences. here groups are presented in a different order compared to other figures.

Response: We appreciate the reviewer's feedback. In response, we have reorganized the order of groups in Figure 3 to ensure consistency with other figures, facilitating a more coherent presentation. Additionally, we have revised the results to provide a clearer description of the observed trends in AF inducibility, emphasizing the impact of the HFD on group differences. The adjusted results now read as follows:

  No significant difference in AF inducibility was observed between ALDH2*2 KI mice after prolonged HFD and WT mice on a ND. Our results suggest that significant differences in AF inducibility are attributed more to the HFD than the ALDH2*2 genotype.

  We thank the reviewer for bringing this to our attention and hope these modifications enhance the clarity of our presentation.

Figure 5 shows an increased expression of HO-1 in aldh2*2 vs. wt mRNA and protein, and increased expression of nrf2 in hfd/aldh2*2 (mRNA and protein); however n=3 and SD quite high.

Response: We appreciate the reviewer's observation. To address concerns about sample size and variability, we have increased the sample size for the mRNA expression and protein experiment in Figure 5 and conducted additional experiments to strengthen the robustness of our data. This adjustment will enhance the reliability and statistical power of our findings. Thank you for bringing this to our attention, and we are committed to improving the quality of our results.

The conclusion is not supported by the presented results. instead, meaningless statements are given.

Response: To address the reviewer's concern, we have revised our conclusion as follows: Our study on ALDH2 deficiency and pacing-induced AF in a murine model with chronic HFD revealed unexpected insights. Contrary to expectations, ALDH2 deficiency did not significantly heighten AF susceptibility in obesity; instead, the activation of the Nrf2/HO-1 pathway suggests an adaptive mechanism. Additionally, our results indicate that significant differences in AF inducibility are primarily attributed to the HFD rather than the ALDH2*2 genotype. These unexpected findings emphasize the need for further investigation into the nuanced relationships between ALDH2 deficiency, obesity, and AF susceptibility. Such insights may guide targeted treatments, especially in populations with a high prevalence of the ALDH2*2 allele.

Although this field of research has interesting content, this study contains methodological and many presentational deficits.

Response: Thank you for your feedback. We acknowledge the methodological and presentational concerns raised. We have carefully reviewed and revised both the methodology and presentation to address these issues. Your insights are valuable, and we are committed to enhancing the overall quality of our study.

Reviewer 4 Report

Comments and Suggestions for Authors

The authors have improved their manuscript significantly and all my concerns have been addressed. Nothing further. 

Comments on the Quality of English Language

Minor editing is needed.

Author Response

We greatly appreciate the recommendations of the reviewer. We have carefully reviewed the manuscript to address any grammatical or stylistic issues.

Reviewer 5 Report

Comments and Suggestions for Authors

The manuscript has been improved and can be considered for publication.

Author Response

(The authors gave the same response as above.)
